# Risk Factors for Respiratory Syncytial Virus Lower Respiratory Tract Infections: Evidence from an Indonesian Cohort

**DOI:** 10.3390/v13020331

**Published:** 2021-02-21

**Authors:** Rowena Crow, Kuswandewi Mutyara, Dwi Agustian, Cissy B. Kartasasmita, Eric A. F. Simões

**Affiliations:** 1Department of Infectious Disease, University of Colorado School of Medicine & Children’s Hospital Colorado, Aurora, CO 80045, USA; rowena.crow@cuanschutz.edu; 2Department of Public Health, Faculty of Medicine, Universitas Padjadjaran-Hasan Sadikin General Hospital, Bandung, Jawa Barat 40161, Indonesia; kuswandewi@gmail.com (K.M.); dwi.agustian@unpad.ac.id (D.A.); 3Center for Collaborative Research on Acute Respiratory Infection, Universitas Padjadjaran-Hasan Sadikin General Hospital, Bandung, Jawa Barat 40161, Indonesia; cbkarta@gmail.com; 4Department of Child Health, Faculty of Medicine, Universitas Padjadjaran-Hasan Sadikin General Hospital, Bandung, Jawa Barat 40161, Indonesia; 5Center for Global Health, Department of Epidemiology, Colorado School of Public Health, Aurora, CO 80045, USA

**Keywords:** epidemiology, home visitation, longitudinal study, air pollution, animal–human interphase

## Abstract

Although risk factors for hospitalization from a respiratory syncytial virus (RSV) are well known, RSV lower respiratory tract infections (LRIs) in the community are much less studied or understood, especially in developing countries. In a prospective, cohort study we studied factors predisposing Indonesian infants and children under 5 years of age to developing RSV LRIs. Subjects were enrolled in two cohorts: a birth cohort and a cross-sectional cohort of children <48 months of age. Subjects were visited weekly at home to identify any LRI, using the World Health Organization’s criteria. RSV etiology was determined through analysis of nasal washings by enzyme immunoassay and polymerase chain reaction. Risk factors for the development of the first documented RSV LRI were identified by multivariate analysis using logistic regression and Cox proportional hazard modeling. Of the 2014 children studied, 999 were enrolled within 30 days of birth. There were 149 first episodes of an RSV. Risk factors for an RSV LRI were poverty (*p* < 0.01), use of kerosene as a cooking fuel (*p* < 0.05), and household ownership of rabbits and chickens (*p* < 0.01). Our findings suggested that in a middle-income country such as Indonesia, with a substantial burden of RSV morbidity and mortality, lower socioeconomic status, environmental air quality, and animal exposure are predisposing factors for developing an RSV LRI.

## 1. Introduction

RSV is the most common cause of acute lower LRI in young children globally as well as an important cause of hospital admissions [1]. It is estimated to have caused 59,600 in-hospital deaths in infants and children under five in 2015, with 99% occurring in developing countries [2]. Estimates of mortality from RSV LRIs in healthy children in developing countries range from 2–7% [3,4], significantly higher than those of both the healthy and high-risk patients from industrialized countries [5]. It is critically important to identify risk factors in developing countries for developing RSV LRIs, and attempt to elucidate possible mechanisms for this higher mortality.

Much of our current understanding regarding the development of RSV LRIs in infants and young children relies upon hospital-based studies with few community-based [6,7,8] and even fewer studies from developing countries [9]. While hospital-based studies have identified risk factors on the severity of an RSV infection––including age, prematurity and underlying immune, pulmonary or cardiac disease [10,11]––these studies are focused on the most severe infections and therefore underestimate the substantial burden of disease within the community. Epidemiological and environmental studies have further identified factors such as crowding, daycare, and smoke exposure, which increase the severity of RSV infections [12].

The objective of this study was to identify demographic, socioeconomic, and environmental factors in Indonesia associated with developing RSV LRIs in children <5 years of age.

## 2. Materials and Methods

Study-Site Description: The study, following a standard World Health Organization (WHO) protocol [13], was conducted at 2 sites located on the northeast outskirts of Bandung City, West Java: Cikutra, a peri-urban community (population: 53,000), and Ujungberung, a semi-rural community (population: 42,000), as described earlier [14]. With the growth of Bandung City and its surroundings, both communities have both urban and rural characteristics with Cikutra having more urban characteristics. Ujungberung is adjacent to a district with a more rural character. The unplanned or loosely planned development at both sites has led to an informal layout of densely packed buildings. Both communities have an outpatient primary health center (Puskesmas) staffed by a physician, nurses, and laboratory technicians. A total of 47 subcommunities were selected from the 2 sites (22 in Cikutra and 25 in Ujungberung), where a total of 102 kaders (female community health workers) conducted weekly home visits to ascertain LRIs in the infants and children.

Patient Recruitment: After obtaining community consent, we conducted a survey of all households in the 47 subcommunities. Enrollment of children <5 years began in February 1999, and they were followed longitudinally for the 28 months of the study or until they reached 59 months of age or dropped out. Subjects for enrollment were randomly selected with approximately 100 children for each of 5 defined age groups: (1) 3–5 months, (2) 6–8 months, (3) 9–11 months, (4) 12–23 months, and (5) 24–48 months. All newborn babies born in both subcommunities were prospectively enrolled and followed up until February 2001. Research manuscripts reporting large datasets that are deposited in a publicly available database should specify where the data have been deposited and provide the relevant accession numbers. If these numbers have not yet been obtained at the time of submission, please state that they will be provided during review. They must be provided prior to publication.

Risk factor survey: After obtaining signed, written informed consent, parents were administered a standardized questionnaire to collect a child’s birth and medical history and household demographic, socioeconomic and environmental factors. In addition, data was collected on household ownership of durable assets (e.g., car, refrigerator, television) and housing characteristics (e.g., type of walls, sanitation).

Monitoring for Development of LRI and Subsequent Management: The presence of LRIs was determined through active surveillance by community kaders conducting weekly home visits to assess for respiratory symptoms. Any child with tachypnea, lower chest wall in-drawing, or audible wheeze was further assessed by a trained local physician, Using the standard World Health Organization case definition, [15] tachypnea was defined as a respiratory rate ≥60/min for infants 0–2 months, ≥50/min for infants 2–11 months, and ≥40/min for infants aged >12 months. If any of the 3 signs of tachypnea, chest wall indrawing or wheezing were present, a nasal wash sample was obtained for RSV testing. All patient management followed WHO guidelines [16].

RSV Testing and Polymerase Chain Reaction (PCR) Methodology: Nasal washes were obtained using standard methodology. Primer and probe sequences for RSV type A (RSV-A) and RSV type B (RSV-B) were as previously described [14].

Statistical Analysis: Data were double-entered into a Microsoft Access database. Statistical analysis was performed using Stata statistical software, version 14.2. Logistic and Cox proportional-hazards regression analyses were performed to find univariate predictors for first documented RSV LRI. Only univariate predictors with *p* < 0.20 significance level were considered for inclusion in both the logistic and the Cox proportional hazards multivariate models. The log-rank test of equality was used to identify categorical variables for Cox proportional hazards multivariate model. For groups of collinear variables, only those with the strongest univariate association were included. A stepwise method was used, starting with the variable with the highest test statistic. The district (peri-urban versus semi-rural) was included in multivariable analyses to account for differences between the settings. Diagnostic tests were performed to test the proportional hazards assumption and residuals were checked. The results are reported as adjusted odds ratios (aOR) and adjusted hazard ratios (aHR) with 95% confidence intervals (95% CI). Four different crowding indices were created based on a ratio of persons to area, (total number of household members/dwelling area, number of siblings <5 years/dwelling area, number of siblings <14 years/dwelling area, number of siblings sharing child’s bedroom/area of bedroom).

A poverty index score was generated from the household ownership of durable assets (e.g., car, refrigerator, television), housing characteristics (e.g., dwelling floor and roof material, toilet facilities), and access to services (e.g., electricity supply, drinking water source). A summary measure was created after applying factor weights to each variable. These factor weights were determined through comparison with the Demographic and Health Survey 2004 national survey data using principal components analysis. In this manner, a poverty index score and the nationally representative wealth quintile was found for each household [17,18,19].

## 3. Results

### 3.1. Subjects

There were 2014 study participants, with 999 enrolled before 1 month of age. Overall, 993 boys (49.3%) were enrolled in the study. Enrollment at the two sites was roughly equal with slightly more subjects enrolled in Cikutra, the peri-urban site (1024; 51%) (Table 1). 

The 2 cohorts showed significant differences in the following baseline factors: the proportion in the newborn cohort compared to the older cohort, hospital born (and hence BCG cicatrix numbers), crowding, smokers in the home, cooking habits, home ventilation, animal ownership, and poverty indices. (Table 1). Of the 149 first episodes of an RSV LRI (141 with complete clinical details), 33 were classified as having severe pneumonia and 3 were classified as having very severe pneumonia using the WHO classification. 

There were no deaths and none of the subjects was hospitalized. The first episodes of an RSV LRI occurred at a mean age of 16.3 months (range: 3.8 to 55.0), with an overall incidence of 7.4%. Our study population was relatively wealthy compared to the general population of Indonesia (64.3% in the wealthy quintile). Residential housing consists predominantly of buildings constructed with cement/stone walls (84%) and with tile or cement floors (50% and 34% respectively). Very few dwellings have dirt or wood floors (2.7% and 2.6% respectively), and 93% of dwellings have a cement-tile roof.

### 3.2. Risk Factors for Developing RSV LRI and All-Cause LRI

#### 3.2.1. Univariate Analyses for RSV LRI

Results from logistic and Cox regressions were similar and showed increased risk for acquisition of RSV LRIs for the following variables: semi-rural residence, non-hospital delivered, less than eighth-grade education for either the mother or father, mother’s occupation (manual or household), household cooking exposure (kerosene or wood as fuel, more than three hours of cooking a day, no ventilation for the kitchen), more than three chickens or rabbits in the house, and lower poverty scores. (Table 2)

#### 3.2.2. Univariate Analyses for All-Cause LRI

Logistic and Cox regressions were repeated for all-cause LRI (Table 3)

#### 3.2.3. Multivariate Analyses for RSV and All-Cause LRI

In a multivariate analysis (Table 4), several of these variables were consistently shown to be independently significant.

An increase in the poverty index score was associated with an increased risk of RSV LRI (aHR = 1.44; 95% CI = 1.17–1.77) (aOR = 1.45; 95% CI = 1.16–1.82) The use of kerosene as a cooking fuel was found to be a risk factor for RSV LRIs (aHR = 2.24; 95% CI = 1.29–3.89) (aOR = 2.16; 95% CI = 1.24–3.76). In addition, having no window ventilation in the kitchen and spending >3 h cooking each day were both associated with increased risk of RSV LRIs (aHR = 1.50; 95% CI = 1.08–2.08) (aOR = 1.61; 95% CI = 1.07–2.45) and (aHR = 1.47; 95% CI = 1.00–2.16) (aOR = 1.61; 95% CI = 1.07–2.45) respectively. Both domestic ownership of rabbits and chickens were associated with an increase in the risk of RSV LRIs. A larger effect was found for the ownership of rabbits (aHR = 2.97; 95% CI = 1.20–7.32) (aOR = 3.04; 95% CI = 1.10–8.44). Owning more than 3 chickens was also statistically significant (aHR = 1.77; 95% CI = 1.15–2.73) (aOR = 1.85; 95% CI = 1.16–2.95).

The ownership of chickens as a risk factor was specific to RSV LRIs. It was not found to be a significant risk factor in all-cause LRIs. The use of kerosene as a cooking fuel remained a significant risk factor in all-cause LRIs though variables related to the exposure level (>3 h cooking, window ventilation) were not significant.

## 4. Discussion

We followed a cohort of children <5 years old living in semi-rural and peri-urban Indonesia to detect RSV LRIs using active surveillance. The incidence of the first episode of an RSV LRI was relatively low at 7.4% overall. There was also a pattern of less severe RSV disease occurring at a later age; The majority of RSV LRI episodes were classified as non-severe while 22% were classified as severe and 2% as very severe pneumonia. As previously reported [14], there was a relatively low incidence of RSV LRIs in the first 6 months of life (5%) with none in the first 3 months. Readers are referred to the previous publication for an extensive discussion on speculation as to why this occurred [14] This observation might be significant in the context of maternal immunization [20] or universal passive prophylaxis with long-acting monoclonal antibodies [21] and whether these newer interventions will reach LMIC in a timely manner [22].

We found that the children living in poorer households were at increased risk of contracting an RSV LRI. The study used an asset-based approach to measure household wealth, an approach first proposed by Filmer and Pritchett [17] that is now widely used to quantify economic status in analyses of health inequalities [18]. This approach was designed to overcome the challenges of measuring income in low- and middle-income countries, where this information is often unavailable, unreliable, and difficult to measure. The ownership of durable goods can be both representative of longer-term wealth and less sensitivity to fluctuations in income.

While it is known that poverty is an important determinant of health, there is limited data on its influence on acute respiratory disease [23], in particular RSV LRIs in young children in middle-income countries. However, poverty as a risk factor is consistent with the higher burden of disease in developing countries compared to industrialized countries.

We also identified the use of kerosene as a cooking fuel as a risk factor in the child’s household for RSV LRIs and all-cause LRIs. In our study population, the vast majority of households (81.2%) use it compared to electricity or gas (16.2%) or wood (2.6%). Although there is sparse data regarding the health risks associated with the use of kerosene stoves, it is known that kerosene use can expose occupants to air contaminants [24]. Its combustion emits many potentially health-damaging pollutants, including particulate matter (PM), carbon monoxide, polycyclic aromatic hydrocarbons and volatile organic compounds, even during normal operation [25]. The fine PM (diameter <2.5 µm) emitted from kerosene combustion enable inhaled particles to be deposited in the deep lung [1,24]. We found that having no window ventilation further increases the risk of RSV LRIs. It would be logical to assume that a lack of ventilation would increase a child’s exposure to air contaminants from kerosene combustion, thereby providing an explanation for this finding.

A previous parental 1-year recall cross sectional study in urban Bangalore, India, [26] found a positive association between cooking with kerosene and bronchitis and phlegm among children aged 0–17 years. However, it did not examine many known risk factors, used nonstandard definitions of outcomes in children, and did not examine LRIs or etiology. Previous studies on biomass fuel use have mainly focused on solid cooking fuels such as wood and dung and have found them to be associated with an increased risk of acute lower respiratory infection (ALRI) [2,27]. Kerosene is often assumed to be “cleaner burning” than biomass fuels and regarded as a “step up the energy ladder” from solid cooking fuels [28]. The use of kerosene for cooking is widespread in many developing countries, replacing the use of biomass fuels, especially in urban populations where electricity and liquefied pressured gas (LPG) can be expensive or unavailable.

Our study also found a novel correlation between the ownership of rabbits and chickens and an increased risk for RSV LRIs. In the study population, 25.5% of households reported raising poultry to eat or sell. Most chickens are allowed to roam freely around the home and surroundings and return to their coup at nighttime. The ownership of rabbits, however, was found to be low (1.2% of households). A previous study of ours—also from rural West Java, Indonesia—also found poultry to be a risk factor for development of H3N2 influenza [29]. Household ownership of livestock such as rabbits and chickens likely leads to exposure to organic dust early in life due to the population’s proximal living with their animals. This exposure may induce an inflammatory response in a child’s lower respiratory system leading to the development of an LRI on exposure to an RSV, but paradoxically, exposure to organic dust early in life might offer protection from a developing a severe LRI. (in this study of over 2000 infants and children followed prospectively for 28 months, most RSV LRIs occurred in older infants with none hospitalized and none dying) [14]. The organic dust, also known as bioaerosols produced by livestock, consists of airborne particulate matter of biological origin such as bacteria, fungi, viruses, high molecular weight allergens, bacterial endotoxins (LPS), mycotoxins, peptidoglycans, β(1→3)-glucans, pollen, and plant fibers. Organic dust is generated by livestock through their feces, litter and feed [30].

While any of the components of organic dust could play a role in RSV-associated LRIs, a prospective study of healthy, full-term infants in Buenos Aires found an association between environmental exposure to LPS and the severity of an RSV infection for infants with the TLR4+/− genotype [31]. This genotype is found in approximately 10% of the population and is associated with increased risk of severe RSV disease [32]. The Buenos Aires study found that for this subgroup of infants with the TLR4 polymorphism, exposure to higher levels of LPS, due to being from a low-SES, rural region, provided a protective effect on the severity of RSV bronchiolitis even after adjusting for other risk factors. This evidence of a gene-environment interaction adds further complexity to understanding the effect of environmental factors and air quality on immune responses and the risk of developing an RSV LRI. The authors of the Buenos Aires study theorize that the TLR4 polymorphism in infants with the TLR4+/− genotype enhances the production of proinflammatory cytokines during an RSV infection. LPS exposure may modulate this innate immune response, suppressing the expression of TLR4 and reducing the severity of an RSV LRI.

However, the effect of bacterial endotoxin on the respiratory system is likely dose-specific with different exposure levels leading to a different innate and adaptive immune responses [33]. The innate immunity senses pathogens and through secretion of cytokines deflects the adaptive immunity into a Th1/Th2/Th17 response. It has been shown that a high-dose endotoxin challenge induces airway neutrophilia and a decrease in phagocytosis by airway monocytes, macrophages, and neutrophils [32]. However, a low-dose LPS challenge primes phagocyte function without inducing airway neutrophilia, appearing to skew airway inflammation in a Th2 direction [34]. The dust generated by poultry contains substantial amounts of endotoxins [3,35].

In our study we cannot comment on the impact of LPS on the severity of disease, and we also did not examine TLR4 polymorphisms, but it should be noted that none of the children in our study died or were hospitalized and given that TLR4 polymorphisms are stable in different populations, rural populations in Indonesia will have a high exposure to LPS, this might explain the development of mild RSV LRIs in these children.

We suspect that the level and type of LPS exposure, gene-environment interactions and the functionally distinct immune responses of infants are at play in our population, and it includes interaction with poultry.

All crowding indices showed a trend of increasing risk of RSV LRIs but none of them reached statistical significance. Low birth weight, male sex and multiple births showed a trend of increasing RSV LRI risk but did not approach borderline statistical significance. These factors have previously been reported in the literature as being significantly associated with the risk of severe LRIs or RSV LRIs in studies in industrialized nations.

A similar surveillance study in a rural Kenyan community [9] found low socioeconomic status to be a risk factor for progression from a mild RSV infection to an LRI. They also identified crowding and malnutrition as risk factors and found higher levels of education of the primary caretaker to be protective.

Limitations to our study include the possibility of multicollinearity due to the type of cooking fuel used in households being related to their poverty index score, as kerosene users are likely to reside in lower socioeconomic neighborhoods than LPG/electricity users [28]. Both measures can be seen as proxies for socioeconomic status though the Buenos Aires study [31] suggests that low SES and high kerosene exposure might correlate with lower SES and rural residence. Furthermore, there is a likely correlation between household animal ownership, living in a more rural setting, with a lower socioeconomic status, which could introduce bias due to confounding. It is worth noting a study that showed home endotoxin exposure may independently increase the risk of wheezing during the first year of life [36]. If infants and children residing in households with chickens and rabbits are more likely to have wheezed from exposure to endotoxins, they would have been more likely to be assessed in our study for an LRI, and a nasal swab would have been obtained because wheezing is one of our 3 diagnostic criteria. This could have led to the oversampling of infants residing in households with chickens and rabbits, leading to a potential bias in our findings. Finally we did not examine samples for other viruses as was done in the PERCH [37], EPIC [38] and GABRIEL [39] studies, but in those studies, where an RSV was the predominant pathogen, coinfections with other viruses (except rhinovirus) were uncommon.

Our study demonstrated that lower socioeconomic status is a risk factor for an RSV LRI in children <5 years of age. The use of kerosene as a cooking fuel as opposed to the use of electricity or gas was also found to be a risk factor. The increased risk due to no kitchen window ventilation and >3 h spent cooking each day demonstrated a dose response mechanism. Household ownership of rabbits and chickens were also found to be risk factors.

This study shows that lower socioeconomic status, environmental air quality and animal exposure are predisposing factors for developing an RSV LRI. In a middle-income country such as Indonesia, with a substantial burden of RSV morbidity and mortality, we recommend conducting a similar study in another area to validate these findings and further investigate the gene-environment interaction, including distinct immune responses of infants for the risk of developing an RSV LRI.

## Figures and Tables

**Table 1 viruses-13-00331-t001:** Characteristics of Infants and Children by Age at Recruitment.

Characteristic	Total N = 2014 n (%)	Child Recruited > 30d N = 1015 n (%)	Newborn Recruited ≤ 30d N = 999 n (%)	*P **
Multiple birth	22 (1.1)	9 (0.9)	13 (1.3)	0.371
Mean age entering study (months)	8.9 (SD = 12.6)	17.6 (SD = 12.7)	0.0 (SD = 0.1)	<0.001
Mean age at first RSV LRI (months)	16.4 (SD = 9.2)	19.5 (SD = 10.2)	12.1 (SD = 4.9)	<0.001
Birthweight				
>2500g	1811 (89.9)	916 (90.2)	895 (89.6)	0.597
1500–2500g	197 (9.8)	95 (9.4)	102 (10.2)	
≤1500g	6 (0.3)	4 (0.4)	2 (0.2)	
Not born in hospital	1399 (77.2)	824 (81.3)	575 (72.1)	<0.001
BCG vaccination	1531 (76.4)	969 (95.5)	562 (56.8)	<0.001
Cicatrix	1198 (60.1)	890 (87.7)	308 (31.4)	<0.001
Multiparity of mother	1189 (59.0)	578 (56.9)	611 (61.2)	0.054
Atopic family member	494 (24.5)	236 (23.3)	258 (25.8)	0.179
Occupation of father				
Admin/Enterprise	918 (53.8)	459 (51.7)	459 (56.1)	0.067
Manual	788 (46.2)	429 (48.3)	359 (43.9)	
Occupation of mother				
Admin/Enterprise	209 (10.7)	109 (11.0)	100 (10.3)	0.871
Manual	200 (10.2)	103 (10.4)	97 (10.0)	
Household	1553 (79.2)	783 (78.7)	770 (79.6)	
Education of father >8th grade	1530 (76.0)	758 (74.7)	772 (77.3)	0.173
Education of mother >8th grade	1319 (65.5)	652 (64.2)	667 (66.8)	0.232
Other children <5 years	530 (26.3)	257 (25.3)	273 (27.3)	0.306
Children 5–14 years	1134 (56.3)	590 (58.1)	544 (54.5)	0.097
District				
Semi-rural Ujungberung	969 (48.1)	518 (51.0)	451 (45.1)	0.008
Peri-urban Cikutra	1045 (51.9)	497 (49.0)	548 (54.9)	
Smokers in home	1718 (85.3)	849 (83.6)	869 (87.0)	0.034
Fuel used for cooking				
Electricity/gas	323 (16.2)	161 (16.0)	162 (16.3)	0.935
Kerosene	1623 (81.2)	818 (81.5)	805 (81.0)	
Wood	52 (2.6)	25 (2.5)	27 (2.7)	
>3 h spent cooking each day	342(17.0)	238 (23.4)	104 (10.4)	<0.001
Kitchen has no window ventilation	662 (32.9)	386 (38.0)	276 (27.6)	<0.001
Animal ownership				
Cats	426 (21.2)	315 (31.0)	111 (11.1)	<0.001
Songbirds	534 (26.5)	297 (29.3)	237 (23.7)	0.005
Rabbits	24 (1.2)	15 (1.5)	9 (0.9)	0.233
Chickens	514 (25.5)	290 (28.6)	224 (22.4)	0.002
>3 chickens	214 (10.6)	121 (11.9)	93 (9.3)	0.057
No. persons in child’s bedroom (mean)	3.02 (SD = 0.79)	3.1 (SD = 0.6)	2.9 (SD = 0.9)	<0.001
Crowding indices (person/10 m^2^)				
Total people/dwelling area	2.13 (SD = 2.28)	2.03 (SD = 2.26)	2.20 (SD = 2.30)	0.107
Children <5 years/dwelling area	0.50 (SD = 0.58)	0.49 (SD = 0.60)	0.50 (SD = 0.56)	0.707
Children 5–14 years/dwelling area	0.85 (SD = 1.09)	0.86 (SD = 1.13)	0.80 (SD = 1.05)	0.862
No. persons in child’s bedroom/area	4.72 (SD = 2.89)	4.88 (SD = 2.70)	4.50 (SD = 3.10)	0.008
Housing				
Walls: cement/stone	1704 (84.7)	871 (85.8)	833 (83.5)	0.001
thatch	88 (4.4)	55 (5.4)	33 (3.3)	
wood	56 (2.8)	19 (1.9)	37 (3.7)	
Floors: tile	1020 (50.7)	507 (50.0)	513 (51.5)	0.004
cement	689 (34.2)	331 (32.6)	358 (35.9)	
wood	53 (2.6)	37 (3.6)	16 (1.6)	
dirt	55 (2.7)	25 (2.5)	30 (3.0)	
plastic carpet	156 (7.8)	95 (9.4)	61 (6.1)	
Roof: cement tile	1872 (93.0)	954 (94.0)	918 (92.0)	0.001
asbestos sheet	56 (2.8)	27 (2.7)	29 (2.9)	
corrugated metal	37 (1.8)	21 (2.1)	16 (1.6)	
wood	30 (1.5)	5 (0.5)	25 (2.5)	
thatch	18 (0.9)	8 (0.8)	10 (1.0)	
National Wealth Quintile				
Poorest	0 (0.0)	0 (0.0)	0 (0.0)	0.637
Poor	34 (1.7)	17 (1.7)	17 (1.7)	
Medium	120 (6.0)	58 (5.7)	62 (6.2)	
Wealthy	1157 (57.5)	572 (56.4)	585 (58.6)	
Wealthiest	703 (35.0)	368 (36.3)	335 (33.5)	
Poverty index score (Indonesia)	−1.77 (SD = 0.96)	−1.81 (SD = 0.97)	−1.70 (SD = 0.95)	0.026

* Chi-square test for categorical variables and t-test for numerical variables were used for statistical test.

**Table 2 viruses-13-00331-t002:** Univariate Analysis of Risk Factors for RSV LRIs.

Characteristic	No RSV N = 1865 n (%)	RSV N = 149 n (%)	HR ^1^ (95% CI)	*P*	OR ^2^ (95% CI)	*P*
Male	911 (48.8)	82 (55.0)	1.26 (0.91–1.74)	0.163	1.28 (0.92–1.79)	0.147
Multiple birth	18 (1.0)	4 (2.7)	2.42 (0.90–6.55)	0.081	2.83 (0.95–8.47)	0.063
Birthweight <2500g	187 (10.0)	16 (10.7)	1.08 (0.65–1.82)	0.759	1.08 (0.63–1.85)	0.781
Not born in hospital	1468 (78.7)	129 (86.6)	1.74 (1.09–2.79)	0.021	1.72 (1.06–2.79)	0.028
BCG vaccination	1419 (76.5)	112 (75.2)	0.81 (0.56–1.18)	0.269	0.93 (0.63–1.37)	0.722
Cicatrix	1108 (60.0)	90 (61.2)	0.91 (0.65–1.27)	0.570	1.05 (0.75–1.49)	0.763
Multiparity of mother	1101 (59.0)	88 (59.1)	0.68 (0.49–0.94)	0.977	1.00 (0.71–1.41)	0.995
Atopic family member	451 (24.2)	43 (28.9)	1.25 (0.88–1.78)	0.221	1.27 (0.88–1.84)	0.203
Occupation of father						
Admin/Enterprise	856 (54.4)	62 (46.6)	reference		reference	
Manual	717 (45.6)	71 (53.4)	1.30 (0.93–1.83)	0.129	1.37 (0.96–1.95)	0.084
Occupation of mother						
Admin/Enterprise	204 (11.2)	5 (3.4)	reference	-	reference	-
Manual	178 (9.8)	22 (15.0)	4.71 (1.78–12.42)	0.002	5.04 (1.87 13.59)	<0.001
Household	1433 (79.0)	120 (81.6)	3.36 (1.37–8.21)	0.008	3.42 (1.38–8.46)	0.008
Education of father >8th grade	1437 (77.1)	93 (62.4)	0.52 (0.37–0.72)	<0.001	0.49 (0.35–0.70)	<0.001
Education of mother >8th grade	1248 (66.9)	71 (47.7)	0.47 (0.34–0.65)	<0.001	0.45 (0.32–0.63)	<0.001
Other children <5 years	1226 (65.7)	100 (67.1)	1.07 (0.76–1.50)	0.710	1.06 (0.75–1.52)	0.733
Children 5–14 years	1042 (55.9)	92 (61.7)	1.25 (0.90 -1.74)	0.188	1.27 (0.90–1.80)	0.165
District						
Semi-rural Ujungberung	882 (47.3)	87 (58.4)	reference	-	reference	-
Peri-urban Cikutra	983 (52.7)	62 (41.6)	0.68 (0.49–0.94)	0.021	0.64 (0.46–0.90)	0.010
Smokers in home	1592 (85.4)	126 (84.6)	0.94 (0.60–1.46)	0.780	0.94 (0.59–1.49)	0.791
Cooking fuel						
Electricity/gas	314 (17.0)	9 (6.1)	reference	-	reference	-
Kerosene	1490 (79.9)	133 (89.3)	3.07 (1.56–6.03)	0.001	3.11 (1.57–6.18)	0.001
Wood	46 (2.5)	6 (4.1)	4.51 (1.61–12.7)	0.004	4.55 (1.55–13.4)	0.006
>3 h cooking per day	308 (16.5)	34 (22.8)	1.41 (0.96–2.07)	0.077	1.49 (1.00–2.23)	0.050
Kitchen has no window ventilation	662 (32.0)	65 (43.6)	1.62 (1.17–2.24)	0.003	1.64 (1.17–2.30)	0.004
Animal ownership						
Cats	385 (20.6)	41 (27.5)	1.35 (0.94–1.94)	0.100	1.46 (1.00–2.13)	0.049
Songbirds	494 (26.5)	40 (26.8)	0.98 (0.68–1.40)	0.899	1.02 (0.70–1.48)	0.924
Rabbits	19 (1.0)	5 (3.4)	3.00 (1.23–7.31)	0.016	3.37 (1.24–9.17)	0.017
Chickens	465 (24.9)	49 (32.9)	1.43 (1.02–2.02)	0.039	1.48 (1.03–2.11)	0.033
>3 chickens	188 (10.1)	26 (17.4)	1.84 (1.21–2.81)	0.005	1.89 (1.20–2.95)	0.006
No. persons in child’s bedroom (mean)	3.0 (SD = 0.8)	3.0 (SD = 0.8)	0.94 (0.76–1.16)	0.556	0.95 (0.77–1.19)	0.674
Crowding indices (person/10 m^2^)						
Total people/dwelling area	2.1 (SD = 2.3)	2.4 (SD = 2.1)	1.06 (1.00–1.12)	0.068	1.05 (0.98–1.12)	0.155
Children <5 years/dwelling area	0.5 (SD = 0.6)	0.6 (SD = 0.5)	1.16 (0.90–1.49)	0.241	1.14 (0.87–1.48)	0.342
Children 5–14 years/dwelling area	0.8 (SD = 1.1)	1.0 (SD = 1.1)	1.10 (0.97–1.25)	0.129	1.09 (0.95–1.24)	0.219
No. people in child’s bedroom/bedroom area	4.7 (SD = 2.9)	4.8 (SD = 2.7)	1.02 (0.96–1.07)	0.545	1.01 (0.96–1.07)	0.635
Poverty score	−1.79 (0.97)	−1.45 (0.75)	1.42 (1.21–1.66)	<0.001	1.43 (1.21–1.70)	<0.001

^1^ HR: Hazard Ratio, estimated by Cox regression model; ^2^ OR: Odds Ratio, estimated by logistic regression model.

**Table 3 viruses-13-00331-t003:** Univariate Analysis of Risk Factors for all-cause LRI.

Characteristic	No LRI N = 1535 n (%)	LRI N = 479 n (%)	HR ^1^ (95% CI)	*P*	OR ^2^ (95% CI)	*P*
Male	732 (47.7)	261 (54.5)	1.27 (1.06–1.51)	0.010	1.31 (1.07–1.61)	0.009
Multiple birth	13 (0.8)	9 (1.9)	1.76 (0.91–3.41)	0.093	2.24 (0.95–5.28)	0.065
Birthweight <2500g	144 (9.4)	59 (12.3)	1.33 (1.01–1.75)	0.040	1.36 (0.98–1.87)	0.063
Not born in hospital	1195 (78.0)	402 (84.3)	1.49 (1.16–1.91)	0.002	1.51 (1.15–1.99)	0.003
BCG vaccination	1168 (76.5)	363 (75.8)	0.91 (0.74–1.12)	0.385	0.96 (0.75–1.22)	0.734
Cicatrix	892 (58.6)	306 (64.6)	1.17 (0.97–1.42)	0.100	1.28 (1.04–1.59)	0.022
Multiparity of mother	897 (58.4)	292 (61.0)	1.11 (0.92–1.33)	0.275	1.11 (0.90–1.37)	0.327
Atopic family member	350 (22.8)	144 (30.1)	1.35 (1.11–1.65)	0.002	1.46 (1.16–1.83)	0.001
Occupation of father						
Admin/Enterprise	725 (56.2)	193 (46.5)	reference		reference	
Manual	566 (43.8)	222 (53.5)	1.40 (1.16–1.70)	0.001	1.47 (1.18–1.84)	<0.001
Occupation of mother						
Admin/Enterprise	182 (12.2)	27 (5.7)	reference	-	reference	-
Manual	145 (9.7)	55 (11.7)	2.32 (1.47–3.68)	<0.001	2.56 (1.54–4.26)	<0.001
Household	1163 (78.1)	390 (82.6)	2.14 (1.45–3.16)	<0.001	2.26 (1.48–3.44)	<0.001
Education of father >8th grade	1197 (78.0)	333 (69.5)	0.68 (0.56–0.83)	<0.001	0.64 (0.51–0.81)	<0.001
Education of mother >8th grade	1055 (68.7)	264 (55.1)	0.60 (0.50–0.72)	<0.001	0.56 (0.45–0.69)	<0.001
Other children <5 years	399 (26.0)	131 (27.3)	1.11 (0.90–1.35)	0.324	1.07 (0.85–1.35)	0.557
Children 5–14 years	834 (54.3)	300 (62.6)	1.36 (1.13–1.64)	0.001	1.41 (1.14–1.74)	<0.001
District						
Semi-rural Ujungberung	698 (45.5)	271 (56.6)	reference	-	reference	-
Peri-urban Cikutra	837 (54.5)	208 (43.4)	0.71 (0.59–0.85)	<0.001	0.64 (0.52–0.79)	<0.001
Smokers in home	1309 (85.3)	409 (85.4)	1.01 (0.78–1.30)	0.952	1.01 (0.75–1.35)	0.953
Cooking fuel						
Electricity/gas	290 (19.0)	33 (6.9)	reference	-	reference	-
Kerosene	1197 (78.6)	426 (89.7)	2.89 (2.03–4.12)	<0.001	3.13 (2.15–4.56)	<0.001
Wood	36 (2.4)	16 (3.4)	3.70 (2.03–6.72)	<0.001	3.91 (1.96–7.79)	<0.001
>3 h cooking per day	257 (16.7)	85 (17.7)	1.06 (0.84–1.34)	0.637	1.07 (0.82–1.41)	0.610
Kitchen has no window ventilation	482 (31.4)	180 (37.6)	1.32 (1.10–1.59)	0.003	1.32 (1.06–1.63)	0.012
Animal ownership						
Cats	311 (20.3)	115 (24.0)	1.22 (0.99–1.50)	0.064	1.24 (0.97–1.59)	0.080
Songbirds	413 (26.9)	121 (25.3)	0.90 (0.74–1.11)	0.338	0.92 (0.73–1.16)	0.477
Rabbits	13 (0.8)	11 (2.3)	2.47 (1.36–4.49)	0.003	2.75 (1.22–6.18)	0.014
Chickens	375 (24.4)	139 (29.0)	1.21 (1.00–1.48)	0.054	1.26 (1.01–1.59)	0.045
>3 chickens	161 (10.5)	53 (11.1)	1.04 (0.78–1.38)	0.786	1.06 (0.76–1.48)	0.721
No. persons in child’s bedroom (mean)	4.7 (SD = 2.9)	4.9 (SD = 2.8)	1.03 (1.01–1.06)	0.017	1.03 (1.00–1.07)	0.064
Crowding indices (person/10 m^2^)						
Total people/dwelling area	2.0 (SD = 2.3)	2.4 (SD = 2.2)	1.06 (1.02–1.09)	0.001	1.07 (1.02–1.12)	0.011
Children <5 years/dwelling area	0.5 (SD = 0.6)	0.6 (SD = 0.6)	1.27 (1.13–1.42)	<0.001	1.37 (1.12–1.68)	0.003
Children 5–14 years/dwelling area	0.8 (SD = 1.1)	1.0 (SD = 1.1)	1.13 (1.06–1.20)	<0.001	1.17 (1.05–1.31)	0.004
No. people in child’s bedroom/bedroom area	4.7(SD = 2.9)	4.9 (SD = 2.8)	1.03 (1.01–1.06)	0.017	1.03 (1.00–1.07)	0.064
Poverty score	−1.1 (SD = 0.7)	−0.95 (SD = 0.7)	1.36 (1.21–1.53)	<0.001	1.39 (1.21–1.60)	<0.001

**Table 4 viruses-13-00331-t004:** Multivariate Analysis of Risk Factors for RSV LRI and all-cause LRI.

	**RSV LRI**				**All-Cause LRI**			
**Characteristic**	**Adjusted HR ^1^** **(95% CI)**	***P***	**Adjusted OR ^2^** **(95% CI)**	***P***	**Adjusted HR ^1^** **(95% CI)**		**Adjusted OR ^2^** **(95% CI)**	***P***
Male	-	-	-	-	1.23 (1.03–1.48)	*0.025*	1.38 (1.10–1.72)	0.005
Age at diagnosis	-	-	0.98 (0.97–1.00)	0.046	*-*	*-*	0.99 (0.98–1.00)	0.049
Atopic family member	-	-	-	-	1.44 (1.18–1.75)	*<0.001*	1.55 (1.22- 1.96)	<0.001
District: Peri-urban versus semi-rural	-	-	-	-	1.32 (1.1–1.59)	*0.003*	1.45 (1.16–1.81)	0.001
Kerosene as cooking fuel	2.24 (1.29–3.89)	0.004	2.16 (1.24–3.76)	0.007	2.10 (1.56–2.82)	*<0.001*	2.37 (1.71–3.29)	<0.001
>3 Hours spent cooking each day	1.47 (1.00–2.16)	0.055	1.61 (1.07–2.45)	0.024	-	*-*	-	-
Kitchen has no window ventilation	1.50 (1.08–2.08)	0.016	1.56 (1.10–2.20)	0.013	-	*-*	-	-
>3 chickens owned	1.77 (1.15–2.73)	0.010	1.85 (1.16–2.95)	0.010	*-*	*-*	-	-
Owns rabbits	2.97 (1.20–7.32)	0.018	3.04 (1.10–8.44)	0.032	2.48 (1.36–4.52)	*0.003*	2.71 (1.19–6.17)	0.017
Poverty Index Score	1.44 (1.17–1.77)	0.001	1.45 (1.16–1.82)	0.001	1.36 (1.21–1.53)	*<0.001*	1.38 (1.19–1.61)	<0.001

^1^ Adjusted HR: Hazard Ratio, estimated by multivariate Cox regression model; ^2^ Adjusted OR: Odds Ratio, estimated by multivariate logistic regression model.

## Data Availability

Contact authors.

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
