# Peer review of "Risk Factors for Respiratory Syncytial Virus Lower Respiratory Tract Infections: Evidence from an Indonesian Cohort"

_viruses, 2021, doi:10.3390/v13020331_

Round 1

Reviewer 1 Report

Review for Viruses:

Risk Factors for Respiratory Syncytial Virus Lower Respiratory Tract Infections: Evidence from an Indonesian Cohort. Crow et al.

In a prospective, cohort study authors studied factors predisposing Indonesian infants and children under 5 years of age to develop RSV LRI. Subjects were enrolled in 2 cohorts, a birth cohort and cross-sectional cohort of children < 48 months of age. They were visited weekly at home to identify any LRI, using the WHO criteria.

Of the 2014 children studied, 999 were enrolled within 30 days of birth. There were 149 first episodes of RSV. Risk factors for RSV LRI using logistic regression and Cox proportional hazard modeling were poverty (p<0.01), use of kerosene as a cooking fuel (p<0.05) and household ownership of rabbits and chickens (p<0.01). In a middle-income country such as Indonesia, with a substantial burden of RSV morbidity and mortality, lower socioeconomic status, environmental air quality and animal exposure were found to be predisposing factors for developing an RSV LRI.

This is an interesting and well done study, conclusions are strongly based on study finding. I have only few remarks.

Major concerns

Discussion: The authors write „As previously reported [20], there was a relatively low incidence of RSV LRI in the first 6 months of life (5%) with no RSV LRI in the first 3 months of life.” This fact should again be discussed shortly in the context of any kind of passive prophylaxis.

Please comment on the predominating diagnosis pneumonia and the not described feature of bronchiolitis.

Minor concerns

Table 1: Birthweight >2500g       1,811 (89.9)       916 (90.2)           895 (89.6)           p=0.597. Row 2 should be written 811 instead of 1,811

Discussion, last para: „Authors should discuss the results and how they can be interpreted from the perspective of previous studies and of the working hypotheses. The findings and their implications should be discussed in the broadest context possible. Future research directions may also be highlighted.” This must be an error and I think this part should be deleted and replaced by a final conclusion.

Author Response

Response to Reviewer 1 Comments

Point 1: Discussion: The authors write „As previously reported [20], there was a relatively low incidence of RSV LRI in the first 6 months of life (5%) with no RSV LRI in the first 3 months of life.” This fact should again be discussed shortly in the context of any kind of passive prophylaxis.

Response 1: We did find a relatively low incidence of RSV LRI in the first 6 months of life (5%) with no RSV LRI in the first 3 months of life. This is a surprising finding given that other studies that have found RSV LRI occurring in the first 6 months of life, with about 50%

of these infections occurring in the first 6 to 12 weeks. Our earlier paper [14] discussed this finding in detail, as outlined below. We did not feel it appropriate to discuss this issue again, but readers are pointed to the article for further perusal.

To quote from that manuscript:

"The most dramatic result of our study was the lack of RSV LRI in infants less than 2 months of age. What may be the explanation for this observation? Most studies36–39 have found that RSV LRI occurs in the first 6 months of life with about 50% of these infections occurring in the first 6 to 12 weeks.36 Recent population-based studies in Africa,38,39 the United States,40 and Holland41 have also shown a low incidence of RSV LRI in children _6 months of age with severe disease occurring mostly in the first 3 months of life. However, our study is different as we found no RSV LRI in the first 2 months of life. Disease was not mild in infants of age 3 to 11 months, as 42% had WHO-defined severe or very severe pneumonia (Table, Supplemental Digital Content 1, http://links.lww.com/INF/A815), which required hospitalization, as per recommendations of WHO. Thus, it was not lack of severe disease in our population that accounted for our findings.

In our study, only 5% of all infections occurred in children less than 6 months and none in those less than 3 months of age. The explanation was not trivial (ie, lack of follow-up). We recruited virtually all newborn babies in the first 3 years into the cohort (almost 1000). Babies were followed carefully from birth onwards by kaders who recruited them within a few days of birth, and it was not lack of obtaining a nasal wash sample. Of the 804 children with LRI, only 17 did not have a nasal wash, which were equally distributed throughout the entire population. It was not that all of these babies were first-born babies with no RSV exposure at home. Only one-third were first-born babies and the average family size in the cohort was between 5.4 and 6.2, with 3.2 and 3.9 being the average number of children per household in Cikutra and Ujungberung, respectively. Social and demographic factors were not an explanation as well because our peri-urban area (Cikutra) included both middle class and upper middle class families, some of whom were physicians, lawyers, business men, and others. In the poorer rural area (Ujungberung), there were significantly higher rates of RSV LRI. In most studies, most children with RSV LRI are actually healthy term babies, which was the majority of our cohort, and the low rate of prematurity was not an explanation. Risk factors that have been shown to contribute to the development of severe RSV LRI include crowding and smoking.36 In this population, _70% of the families have smoking members in the household, and the average family size being 6 is significantly larger than family sizes in Europe and North America where RSV LRI occurs in infants less than 2 months of age. Hence, the lack of risk factors is not an explanation either.

Breastfeeding might have protected against RSV LRI in the first 2 months of life. However, in comparison to the rest of the world, the breastfeeding rates in the study were comparable. Thus, exclusive breastfeeding rates and rates of partial breast-feeding are comparable to those in the rest of Asia and Africa. [37–39] It is possible that we did not sample enough children. In fact, our rates of RSV LRI are 2-fold higher (despite the fact that we had no RSV LRI in infants less than the age of 3 months) in infants <1 year of age than comparable studies from India.[37] These rates (60 RSV LRI per 1000 infants 1 year of age) are about half the rate reported from Kenya,38 but the majority of diseases in that study was in younger infants. High concentrations of maternal RSV antibody in Indonesian infants are probably not the reason either. Our own study [42] has shown that epidemics of RSV in Denmark correlate very well with the increase and fall of maternal antibodies in the cord blood. Indeed, maternal antibody decreases in a periodic manner commensurate with RSV epidemics. The decrease in maternal RSV antibodies precedes the epidemic by about 2 months and follows the peak of the epidemic by 2 months. In this study from Indonesia, the periodicity of the RSV epidemics suggests that this phenomenon occurs in Indonesia too, although we have not examined maternal antibodies. However, the very marked reproducible seasonality of RSV in Bandung appears to mimic the situation in Northern Europe and North America where such a phenomenon has been well described and seems to correlate with maternal antibody. We intend to study maternal neutralizing antibodies to RSV in this population to answer this question. If there are high titers, it may support maternal immunization strategies."

Regarding passive immunoprophylaxis, (palivizumab, the only one in use currently, is very expensive and used mostly and sparingly at that, in high income countries). Once again, we feel it is beyond the scope of this manuscript on risk factors for developing RSV in a low middle income country (LMIC). Thus, passive immunoprophylaxis is not used at all in any of the 2020 world bank list of LMIC and in a very few of the upper middle income countries, it is used very sparingly (eg: Botswana, Brazil, Costa Rica, Namibia and South Africa.) With the advent of newer long acting monoclonal antibodies that are being targeted for birth doses [21], the senior author in a recent publication, has indeed raised the question of whether these will ever reach LMIC infants [22].

Hence, the finding of low rates of severe RSV disease and its relationship to passive immunoprophylaxis, while very topical and important, we feel are best discussed elsewhere, and we have not included this in the current risk factor manuscript , though we have pointed readers to these discussions. Please see added few sentences in text:

Readers are referred to the previous publication for an extensive discussion on speculation as to why this occurred [14] This observation might be significant in the context of maternal immunization [20] or universal passive prophylaxis with long acting monoclonal antibodies [21] and whether these newer interventions will reach LMIC in a timely manner [22].

Point 2: Please comment on the predominating diagnosis pneumonia and the not described feature of bronchiolitis.

Response 2: In this study the WHO classification was used in assessment of an ALRI.

In LMIC, almost exclusively physicians use the WHO classification and this classification does not differentiate between pneumonia, bronchiolitis or asthma. ALRI are classified as non-severe, severe and very severe pneumonia [13-16].

Point 3: Table 1: Birthweight >2500g       1,811 (89.9)       916 (90.2)           895 (89.6)           p=0.597. Row 2 should be written 811 instead of 1,811.

Response 3: 1811 is the correct number, the comma has been removed

Point 4: Discussion, last para: „Authors should discuss the results and how they can be interpreted from the perspective of previous studies and of the working hypotheses. The findings and their implications should be discussed in the broadest context possible. Future research directions may also be highlighted.” This must be an error and I think this part should be deleted and replaced by a final conclusion.

Response 3: We concur with the reviewer and therefore have deleted the last paragraph and added the following:

This study shows that lower socioeconomic status, environmental air quality and animal exposure are predisposing factors for developing an RSV LRI. In a middle-income country such as Indonesia, with a substantial burden of RSV morbidity and mortality, we recommend conducting similar studies in other areas to validate these findings and further investigate the gene-environment interaction, including distinct immune responses of infants for the risk of developing RSV LRI.

Reviewer 2 Report

Summary

The authors summarized a group of environmental risk factors that are critical for RSV infection within an Indonesian cohort, which is helpful to understand the other potential risk that can increase the RSV infected cases. This is a study for small group in some developing countries, which provides many influencing factors that cannot be ignored.

Specific comments

  1. Authors ignored the other infectious factors and individual immunocompromised possibility when they analyzed the environmental risk. These two aspects are necessary and helpful to estimate if the environmental factors are critical for infection within an Indonesian cohort.
  2. Adding the analysis methods and description under each Table.
  3. Please be careful when concluding the findings in the text, since the data described in the manuscript limited at a small group, which cannot represent all cases in developing countries.
  4. The written needs to be improved.

Author Response

Response to Reviewer 2 Comments

Point 1: Authors ignored the other infectious factors and individual immunocompromised possibility when they analyzed the environmental risk. These two aspects are necessary and helpful to estimate if the environmental factors are critical for infection within an Indonesian cohort.

Response 1: While we agree with the reviewer that knowledge of other infectious pathogens would be beneficial in understanding the environmental risk, in this study we did not test for other viruses and therefore we cannot answer the question of interaction with other viruses.

However, in large studies in developing countries [37], of which the senior author was a member of the study group, in most children with severe pneumonia and RSV, RSV was either the only pathogen or the predominant pathogen.

This was confirmed in the EPIC study in the US [38]and the Gabriel [39] study in developing countries. Hence, while we concur that other infections may have influenced the results, it would have had a minor, if any, impact on our findings of the risk factors for development of RSV LRI in this community-based study. We have added this as a limitation to the study. This is the added sentence in the text.

Finally we did not examine samples for other viruses as was done in the PERCH [37], EPIC [38] and GABRIEL [39] studies, but in those studies when RSV was the predominant pathogen and coinfections with other viruses (except rhinovirus) were uncommon.

This was a community-based study in rural and semi-urban Indonesia. As such, we studied 2014 children, who were either recruited at start of study of recruited at birth. It is highly unlikely that this small population would have a significant number of immunocompromised children. Furthermore, there were low rates of HIV in the population at time we did the study. Of note, most babies with severe immunodeficiency (SCID) would typically not survive in a LMIC country. Had this had been a hospital based study this would have certainly been an important consideration.

Point 2:    Adding the analysis methods and description under each Table.

Response 2: The methods and description have been added as footnotes to tables

 Point 3:    Please be careful when concluding the findings in the text, since the data described in the manuscript limited at a small group, which cannot represent all cases in developing countries.

Response 3: We concur with the reviewers and have revised the abstract conclusion and the last paragraph of discussion:

Abstract:

Our findings suggest that in a middle-income country such as Indonesia, with a substantial burden of RSV morbidity and mortality, lower socioeconomic status, environmental air quality and animal exposure are suggested to be predisposing factors for developing an RSV LRI.

Last paragraph:

This study shows that lower socioeconomic status, environmental air quality and animal exposure are predisposing factors for developing an RSV LRI. In a middle-income country such as Indonesia, with a substantial burden of RSV morbidity and mortality, we recommend conducting similar study in other area to validate these findings and further investigate the gene-environment interaction, including distinct immune responses of infants for the risk of developing RSV LRI for future study.

Point 4:    The written needs to be improved.

 Response 4: The manuscript has been edited for grammar and clarity

Round 2

Reviewer 2 Report

Authors addressed all the comments appropriately.